# A Dynamic Culture Method to Produce Ovarian Cancer Spheroids under Physiologically-Relevant Shear Stress

**DOI:** 10.3390/cells7120277

**Published:** 2018-12-19

**Authors:** Timothy Masiello, Atul Dhall, L. P. Madhubhani Hemachandra, Natalya Tokranova, J. Andres Melendez, James Castracane

**Affiliations:** Colleges of Nanoscale Science and Engineering, SUNY Polytechnic Institute, Albany, NY 12203, USA; adhall@sunypoly.edu (A.D.); lhemachandra@sunypoly.edu (L.P.M.H.); ntokranova@sunypoly.edu (N.T.); jmelendez@sunypoly.edu (J.A.M.); jim.castracane@gmail.com (J.C.)

**Keywords:** ovarian, cancer, peritoneal, spheroid, shear, shaker, fluid, dynamic

## Abstract

The transcoelomic metastasis pathway is an alternative to traditional lymphatic/hematogenic metastasis. It is most frequently observed in ovarian cancer, though it has been documented in colon and gastric cancers as well. In transcoelomic metastasis, primary tumor cells are released into the abdominal cavity and form cell aggregates known as spheroids. These spheroids travel through the peritoneal fluid and implant at secondary sites, leading to the formation of new tumor lesions in the peritoneal lining and the organs in the cavity. Models of this process that incorporate the fluid shear stress (FSS) experienced by these spheroids are few, and most have not been fully characterized. Proposed herein is the adaption of a known dynamic cell culture system, the orbital shaker, to create an environment with physiologically-relevant FSS for spheroid formation. Experimental conditions (rotation speed, well size and cell density) were optimized to achieve physiologically-relevant FSS while facilitating the formation of spheroids that are also of a physiologically-relevant size. The FSS improves the roundness and size consistency of spheroids versus equivalent static methods and are even comparable to established high-throughput arrays, while maintaining nearly equivalent viability. This effect was seen in both highly metastatic and modestly metastatic cell lines. The spheroids generated using this technique were fully amenable to functional assays and will allow for better characterization of FSS’s effects on metastatic behavior and serve as a drug screening platform. This model can also be built upon in the future by adding more aspects of the peritoneal microenvironment, further enhancing its in vivo relevance.

## 1. Introduction

Ovarian cancer (OC) is the deadliest gynecological cancer, expected to cause over 14,000 deaths in 2018, with a five-year survival rate that can be as low as 30% [1]. OC’s high fatality rate arises from both difficulties in diagnosis and treatment. About 90% of ovarian cancers are epithelial in origin (Epithelial Ovarian Cancer—EOC) [2]. There are four major subtypes of EOC, two of which attract strong interest. First, and most common, is high-grade serous carcinoma (HGSC, ~70% of clinical cases), which is believed to originate in the fallopian tubes via serous tubal intraepithelial carcinoma (STIC) lesions [3]. The second is clear cell carcinoma (CCC), which accounts for about 10% of clinical cases [4], and closer to 25% for Asian women. While not as common, clear cell carcinoma has attracted significant attention due to its highly aggressive nature compared to other subtypes [5]. In both cases, however, the symptoms are typically mild digestive complications and/or pelvic pain, among others, that can be easily mistaken for other conditions [6]. If caught early, cytoreductive surgery and follow-up chemotherapy have high success rates; however, unfortunately, over 60% of patients are diagnosed at stage III or IV [7], and the prognosis for those patients is considerably worse [8].

While capable of lymphatic/hematogenic dissemination via the vasculature [9], most clinical cases of ovarian cancer metastasis involve the transcoelomic pathway [10] (Figure 1). In transcoelomic metastasis, a mode also utilized by colon and gastric cancer [11,12], cells are shed from the primary tumor into the peritoneal cavity, subsequently forming cell aggregates called spheroids [13]. The size of spheroids typically ranges from 100–700 µm in diameter [14]. These spheroids follow the fluidic currents of the abdominal cavity until they implant at a secondary site. This is often on the omentum, a fat pad covering the bowel and abdominal cavity [15]. The spheroids then clear the outer layer of mesothelial cells [16], attach, disaggregate and invade [17], forming the basis for the growth of a secondary tumor. Highly chemoresistant and boasting a complex microenvironment of nutrient/oxygen gradients and cell properties [18], the use of spheroids for ovarian cancer studies has become commonplace alongside traditional monolayer culture over the past twenty years [19].

Spheroids form spontaneously upon cell culture conditions in a low-attachment environment. Two of the most common methods are the hanging drop method [20], where drops of cells in suspension are dangled off the top of a Petri dish lid, and liquid overlay, where the surface of tissue culture plates are coated with a neutral compound such as poly(2-hydroxyethyl methacrylate) (poly-HEMA) or agarose to prevent cell adhesion [21], making them ultra-low attachment (ULA). Both assays have been adapted into commercially-available high-throughput options [22,23]. However, these methods have a key limitation: they are often static, not accounting for the dynamic fluid environment of the peritoneal cavity. It has been demonstrated that the addition of low amounts of fluid shear stress (FSS) alters the properties of spheroids, such as increasing the expression of cancer stem cell markers and chemoresistance [24]. There is always a need to improve the in vivo relevance of in vitro models and the addition of this variable could improve ovarian cancer studies of invasion, epithelial-to-mesenchymal transition (EMT) and chemoresistance.

FSS can be applied to spheroids in many ways. Very low levels of FSS, commonly ranging from 2 to below 0.1 dyne/cm^2^ (which is equivalent to decipascals, dPa), can be achieved using microfluidic channels coupled to a syringe pump with low flow rates [24,25]. However, a method that is a better special mimic of the roughly circular and wide movement within the peritoneal space is the rotating movement of a bioreactor [26]. Bioreactors are popular tools for large-scale suspension culture of both mammalian and microbial cells, namely for bioproduction [27]. However, the greatest drawback to these systems is the concern about harmful shear stress, namely what is produced by the typically turbulent movement of the stirring apparatus [28]. An example where this apparent weakness is exploited for experimental benefit, however, is with orbital shakers. Commonly used for deliberately inflicting FSS on adherent cells [29,30], shaker rotation speeds can be as high as 300 rpm. Assuming all other conditions are constant, the primary variables to modulate FSS are orbital diameter and rotation speed [31]. Maximum FSS generated on orbital shakers can, depending on the balance of the preceding variables, be over 10 dynes/cm^2^ or as low as <1 dyne/cm^2^. For suspension culture, rotation speeds of 70–110 rpm are common [32,33], generally producing a maximum FSS of around 0.5–1 dyne/cm^2^. This value falls within the physiological range of shear expected in the abdominal cavity where peritoneal metastasis occurs and what has been previously investigated [34,35,36].

Here we report the first characterization and optimization of an orbital shaker/rotating plate for producing ovarian cancer spheroids under physiologically-relevant FSS, and directly comparing to the morphology and viability of those produced by common static methods. With properly-tuned culture conditions the use of a shaker allows production of spheroids with rounder, more consistent morphology versus equivalent static plates. While this method lacks the individualized control over each spheroid that a high-throughput method would offer, it compensates by having a more flexible setup and it is easier to conduct varied, long-term culture experiments that would be time-consuming and challenging to perform otherwise. The spheroids here show consistently high viability and are easily handled for functional assays. The establishment of this method will shed new light on ways to produce spheroids in vitro closer to those in vivo, improving tests for parameters such as invasion, tumor dormancy and drug resistance.

## 2. Materials and Methods

Media and supplements were purchased from Corning, Inc (Corning, NY, USA). Unless otherwise noted, all other chemicals were purchased from Sigma-Aldrich (St. Louis, MO, USA).

### 2.1. COMSOL Multiphysics^®^ Simulations

The Computational Fluid Dynamics (CFD) module was used for shear studies, specifically single-phase flow (spf) studies under laminar flow conditions. The well sizes, volumes, etc. are listed in Appendix A. For incompressible, single phase flow under laminar conditions, and following the Navier-Stokes equation, the velocity of the fluid was determined using the following equations, with FSS calculated within the program:(1)$$\rho \left(u\cdot \nabla \right)u=\nabla \cdot \left[-\rho I+\mu \left(\nabla u+{\left(\nabla u\right)}^{T}\right)\right]+F+\rho g$$
(2)ρ∇·(u)=0
where *ρ* = Fluid density, *g* = Acceleration due to gravity, *u* = Flow velocity, *μ* = Dynamic viscosity, *F* = External force. For consistency, in each case the well was assumed to be offset right next to the axis of rotation, creating an area of maximum shear stress on the left side.

### 2.2. Cell Culture

ES-2 (ATCC CRL-1978) and OVCA420 cells were provided by Dr. Nadine Hempel [37] and grown in McCoy’s 5A and RPMI-1640 media, respectively, both supplemented with 10% fetal bovine serum and 1% penicillin/streptomycin. Cells were incubated at 37 °C in a humidified incubator with a 5% CO_2_ atmosphere. Passaging via trypsinization was performed every 2–3 days at 70–80% confluence.

### 2.3. Preparation of ULA Plates

A 120 mg/mL stock solution of poly(2-hydroxyethyl methacrylate) (poly-HEMA) was prepared in 95% ethanol. The 1:10 working solution was prepared (also in 95% ethanol), placed on culture wells and allowed to dry completely twice to double-coat. Before cell culture, wells were washed twice with 1× phosphate-buffered saline (PBS) and sterilized under ultraviolet (UV) for 30 min.

### 2.4. Spheroid Culture

Passaged cells were counted on a TC-10 cell counter, diluted to the appropriate working concentration, and then 3 mL was placed in each well of the six well plate (WP), 1.5 mL for the 12 WP and 1 mL for 24 well plates. FSS was generated using a SciLogex Micro Mixer Plate (orbital radius: 4.5 mm) for 72–168 h under standard culture conditions. Concurrent static ULA cultures were grown adjacent to the shaker. For round-bottom 96 well plates, 1000 and 2000 cells were seeded per well for ES-2 and OVCA420, respectively. For culture times longer than 72 h, the media was changed at 72 h and every 48 h thereafter.

### 2.5. Microscopy

All spheroids were handled with clipped pipet tips to minimize spheroid damage. All images were taken on an eVOSfl AMG LED-based fluorescent microscope (Thermo Fisher, Waltham, MA, USA). The images were taken under brightfield conditions at either 4× or 10× magnification.

### 2.6. Spheroid Characterization

Image analysis was done in ImageJ (version 1.52a, National Institutes of Health, Bethesda, MD, USA), with the spheroid major and minor axes and roundness, circularity and solidity (RCS) all calculated within the program. The diameter was determined to be the average of the major and minor axes. The spheroid formation was considered unsuccessful when there were no spheroids present greater than 100 µm in diameter, or when the spheroid quantity was less than five per well. The presence of a spheroid of greater than 600 µm indicated that the conditions were not feasible for production of appropriately sized spheroids for in vivo relevance.

### 2.7. Scanning Electron Microscopy (SEM)

Spheroids were fixed overnight in a mixture of 2% paraformaldehyde (PFA) and 2% glutaraldehyde. Samples were dehydrated using ethanol gradations and critical point drying was achieved with increasing concentrations of hexamethyldisilazane (HMDS). Samples were prepared on glass coverslips and attached with mounting media. Sputter-coating with gold/palladium was done on a Denton Vacuum Desk IV (Denton Vacuum, Moorestown, NJ, USA) and images were taken on a Hitachi S-4800 SEM (Chiyoda, Tokyo, Japan) at 5.0 kV and a magnification of 250×.

### 2.8. Live/Dead Staining of Spheroids

Spheroids were stained with prodidium iodide (PI) and Hoescht for 30 min at 20 μg/mL and 0.1 mM final concentrations, respectively, and imaged using Texas Red (535/617 nm) and DAPI (360/460 nm) filters on the EVOS.

### 2.9. Spheroid Viability

The MTT assay was used to assess spheroid viability. One spheroid was placed in each well in the presence of serum-free media and 0.05mg/mL MTT reagent and incubated for 6 h. The formazan dye was solubilized with dimethyl sulfoxide (DMSO) for 30 min and the absorbance at 540 nm was measured on a SpectraMax^®^ Paradigm^®^ Multi-Mode Detection Platform (https://www.moleculardevices.com/en/assets/tutorials-videos/br/spectramax-paradigm-multi-mode-microplate-reader#gref). Readings were normalized to the protein concentration as determined by the BCA assay.

### 2.10. Data Analysis

All image measurements, including diameter and RCS values, were done in ImageJ. Shaker RCS values were normalized to average round-bottom 96 WP spheroid RCS values. Results are given as the mean ± SEM from at least three different experiments. Graphing and statistical analysis were done in Graphpad Prism^®^ 8 (Graphpad, San Diego, CA, USA), with a one-way ANOVA and Tukey post-hoc analysis for spheroid diameter comparisons, a two-way ANOVA with Bonferroni post-hoc analysis for MTT data and a two-way ANOVA with Tukey post-hoc analysis for RCS measurements. In all graphs, a *p* value of <0.0001 corresponds to ****, <0.001 corresponds to ***, >0.01 corresponds to **, and >0.05 corresponds to *.

## 3. Results

### 3.1. Simulation of FSS in Well Plates on an Orbital Shaker

The orbital shaker used here can accommodate 6–96 well plates (WPs) as well as Petri dishes. The fluid environment within the wells on the shaker was simulated using COMSOL Multiphysics^®^ 5.4 using different well sizes and rotation speeds. Simulation results are shown in Figure 2. The initially chosen speed based on common speeds in the literature was 100 rpm. The center of rotation was set at the leftmost side of the circumference for all wells/dishes. The simulations generated values of dPa, which are equivalent to dynes/cm^2^, will be the unit used here. The maximum FSS values were, on average, below 0.5 dyne/cm^2^. Smaller wells tended to have pockets of higher FSS close to the axis of rotation while the larger wells of six WP and petri dishes show more diffuse FSS. The lowest values, below 0.1 dyne/cm^2^, were seen for the 48 and 96 WPs, being one and two orders of magnitude below the others, respectively. Together, these simulations illustrate not only the physiological relevance of the shear stress generated in the plates used but also why it is difficult to use the small-well, high-throughput plates for this method. The fluid is so constrained that the generated FSS falls lower than what would be expected in vivo.

Based on these results, 48 and 96 WPs were excluded due to their minimal fluid movement. With the remaining options, the optimal choice for experiments would balance the throughput level with generating the most appropriate shear stress. To maximize throughput, the Petri dishes were also excluded. For the next step, experiments were conducted with 6, 12 and 24 WPs. Further simulations were run with these three options and the maximum shear stress is plotted versus rotating speed in Figure 3. To achieve a low, but appreciable, shear stress value, the upper and lower limits for our experiments were set at 1.0 and 0.25 dynes/cm^2^ with a general target of about 0.5 dynes/cm^2^.

### 3.2. Optimization of Well Size and Rotation Speed

The initial rotation conditions tested were 6, 12 and 24 WP at 100 rpm with ES-2 cells at a cell density of 100,000 cells/mL. The results showed the spheroids in wells in both 12 and 24 WPs to be very large (1 mm in diameter), far larger than the expected in vivo size (Figure 4E,F). The quantity and size of spheroids was improved in 6 WP (Figure 4D); however, there were still noticeable quantities of irregular, misshapen spheroids. Therefore, the speed was increased to 120 rpm, improving the spheroid quantity in the 12 WP (Figure 4H), yielding 30 or more spheroids per well, all approximately 200–300 µm in diameter. This is a marked improvement from the random distribution of spheroid size and morphology in static 12 well plates, with most of the spheroids there being elongated and irregular (Figure 4B). Additionally, there was slight improvement in the 6 WP (Figure 4G), though the results were not significantly improved in the 24 WP between the static and rotating plates at either 100 or 120 rpm (Figure 4F,I). Finally, the speed was raised to 140 rpm, pushing the shear stress just above the general boundaries set in Figure 3. The spheroid yield and general morphology was unchanged in all 3 plates, with the 24 WP still yielding only 1–2 large spheroids per well (Figure 4L). In all cases, cell incorporation into spheroids was homogenous and thorough.

The above conditions were extended to the OVCA420 cell line at 100,000 cells/mL, with 100 rpm being excluded based on the results of ES-2. This time, 120 rpm was not successful with spheroid formation, yielding 1–2 large spheroids in both 12 and 24 WPs (Figure 5E,F) with no significant spheroid formation in the 6 WP. At 140 rpm the results were improved in the 12 WP but not the 24 WP (Figure 5H,I), and while the spheroids formed in the 6 WP were the appropriate size and morphology the yield was low (fewer than ten per well) compared to the dozens per well in the 12 WP (Figure 5G). The FSS distributions at 120 and 140 rpm are shown in Appendix A.

For further experiments and analysis, the 24 WP were excluded due to their inability to yield spheroids of an appropriate size and quantity. The 6 WP were also excluded; not only had they been minimally effective for OVCA420, but the 12 WP offered higher throughput for only a slight increase in FSS. Therefore, all shaker experiments were conducted with 12 WP. A summary of the parameters tested for this optimization can be seen in Table 1. For each cell line the optimal conditions maximized the well number while minimizing rotation speed. Additionally, theyyielded a high spheroid quantity with morphology that resembles the ULA controls. For each cell line, the condition that fulfills all those conditions is highlighted in blue.

The three-dimensional (3D) nature of shaker-produced spheroids was confirmed via SEM imaging. Both ES-2 and OVCA420 shaker spheroids show clear 3D structures in Figure 6, confirming that the shaker can successfully produce 3D spheroids just as the static methods can.

### 3.3. Effects of Cell Density on Spheroid Size and Morphology

With successful spheroid production at 100,000 cells/mL, different cell densities of ES-2 and OVCA420 were tested to see any effects on spheroid size/morphology, and to find a lower limit of cell density where spheroids would not form. Different cell densities can be used to mimic tumors of different sizes. Additionally, the use of as few cells as possible will make this method more compatible with using primary cells, where generating large quantities can be challenging.

The morphology of the spheroids formed in the 12 WP was quantified using the roundness, circularity and solidity (RCS) measurements. All these parameters were normalized to the corresponding average values for spheroids formed in a round-bottom 96 WP, a common high-throughput method [37]. The RCS values obtained for 100,000 cells/mL come within 0.95 (for ES-2) and 0.9 (for OVCA420), showing high correlation with the static method.

For ES-2, spheroid morphology remains round and consistent, and the spheroids numerous down to 25,000 cells/mL (shown in Figure 7A–C). Below this density, spheroid formation is minimal. The RCS value remains high for the lower cell densities, even showing improvement at 25,000 cells/mL (Figure 7D). Moreover, the average spheroid size increased with decreasing cell density (Figure 7E). Additional representative images are shown in Appendix A.

The RCS trends were partially reproduced for OVCA420. They too can produce consistent spheroids down to 25,000 cells/mL (Figure 8A–C). However, there was a decrease in the roundness and circularity, as well as a decrease in spheroid size. This is most significant going from 100,000 to 25,000 cells/mL (Figure 8D,E). Additional representative images are shown in Appendix A.

### 3.4. Spheroid Viability

A reduction in viability from shaking/stirred cell culture is a constant concern [38], so the viability of spheroids was assessed under FSS. This was performed by staining spheroids grown for 72 h (Shaker at 120 rpm and 100,000 cells/mL for 72 h) and a round-bottom 96 WP with propidium iodide (PI) to visualize necrotic (dead) cells, and a Hoechst stain. PI staining of static and shaking ES-2 spheroids were similar with only a slight qualitative decrease under shaking conditions (Figure 8A,B). However, some level of spheroid necrosis would be expected in larger spheroids due to more limited oxygen and nutrient transport; thus, a second method was utilized to verify these results. Spheroid viability was also quantified using the MTT assay, where shaking enhanced the viability of ES-2 spheroids while there was no significant difference in MTT activity between static and shaking conditions for OVCA420 spheroids (Figure 9E). Thus, the viability of shaker-grown spheroids was not decreased following 72 h of FSS, a time range that is common for spheroid-based experiments in vitro.

### 3.5. Long Term Culture

ES-2 spheroids were assessed for how well they would maintain their morphology and viability in long-term culture. As it is unclear how long, on average, it takes a spheroid to reach a secondary site in vivo, the length of time spheroids are exposed to FSS could vary. This was tested for the ES-2, the more proliferative of the two cell lines, with media changes at 72 h and every 48 h up to 168 h. The roundness, circularity and solidity remained consistent until 168 h (Figure 10G), and while there was an increasingly evident necrotic core in the shaker spheroids going from 72 to 120 to 168 h, this coincided with an increase in size (Figure 10H,I). The spheroids were continuing to proliferate and cells near the spheroid periphery remained viable. Representative brightfield images of these spheroids are shown in Appendix A.

## 4. Discussion

Much about transcoelomic metastasis remains incompletely understood, and this lack of understanding hinders improving treatments for ovarian cancer. There is an ongoing need for improved in vitro models of ovarian cancer and one way to improve their in vivo relevance is to incorporate the FSS associated with transcoelomic metastasis with spheroid culture to better mimic the peritoneal cavity [39]. Ideally, such a method should be simple, accessible and versatile so that it can be applied to any of the numerous facets of metastasis-based studies.

The proposed model here uses an orbital shaker to form spheroids under physiologically—relevant FSS. The shear stress was modeled using COMSOL Multiphysics^®^ with differing cell culture well diameters and rotation speeds, showing the levels of FSS were physiologically relevant. Once base conditions were set for cell-based experiments, it was found that well size, rotation speed cell line and cell density all played roles in determining the size, morphology and viability of spheroids. The highly aggressive ES-2 and modestly aggressive OVCA420 cell lines formed spheroids of comparable morphology and viability to a high-throughput round-bottom 96 WP at a variety of densities with high reproducibility. The 3D nature of the spheroids was confirmed with SEM. Both PI/Hoescht staining and MTT assay indicate that the spheroid viability was not reduced by FSS and that ES-2 spheroids could be maintained in shaker culture with consistent morphology and viability for approximately 7 days.

The behavior of these cell lines suggests that ES-2 adapted to the rotating environment more effectively in terms of spheroid formation efficiency as compared to OVCA420. Prompt survival and adaption to FSS is expected to increase the speed and efficacy of metastasis; thus, it stands to reason that those cells that adapt more effectively are more likely to be highly metastatic. ES-2 is known to be an extremely metastatic cell line within a highly aggressive subtype of ovarian cancer [37]; thus, its high RCS values and enhanced viability under FSS are appropriate for its background. In contrast, OVCA420, known to be modestly metastatic [37], survives FSS but shows lower RCS values and no improvement in viability. Additionally, OVCA420’s optimal spheroid-forming conditions were far more limited than those for ES-2, both from rotation speed/well size and cell density standpoints. This trend of increased adaptation by more metastatic cells was also observed by Hyler, et al. with mouse cell lines [26].

More work is needed to determine the exact reasons for these differences between cell lines, however, a possible explanation is the mesenchymal nature of the two cell lines. ES-2 is highly mesenchymal [40], which that may lend itself to more effective adaption. Supporting this, Rizvi, et al. showed an increase in EMT markers in ovarian cancer cells following FSS exposure [36]. Other possible factors may be cancer stem cell markers (CSC) or even the differences between HGSC and CCC. An increase in CSC markers has been associated with FSS in previous studies [24]. Work with additional cell lines together with comprehensive genomic/proteomic profiling will shed light on the factors that control spheroid formation.

The future applications of this platform are extensive. The reproducibility and consistency of the shaker spheroids makes them ideal for invasion assays and drug testing. Furthermore, shakers tend to be inexpensive (depending on the manufacturer/vendor and exact specifications), and with no need for unique media supplements or extensively modified protocols, this is a method with high accessibility that could be easily incorporated into any lab equipped for cell culture practices. There is also vast room for continuing to layer on more in vivo factors. Future experiments with this platform will include validation with primary cell lines. Additionally, ascites fluid isolated from patients will be used as a supplement for cell culture media [41]. Furthermore, the spheroids will be co-cultured with fibroblasts and mesothelial cells grown in a matrix [42]. These factors will further enhance the strength of this model and increase our understanding of ovarian cancer metastasis. Finally, this model can be easily adapted for other cancer types that utilize transcoelomic metastasis.

## 5. Conclusions

In vitro models of transcoelomic metastasis can be improved by the incorporation of fluid shear stress (FSS) during spheroid formation and growth. The present work has characterized the use of an orbital shaker/rotating plate to grow ovarian cancer spheroids under FSS to use as a platform for studies of FSS’s effect on invasion and chemoresistance. The data showed that the shaker can produce FSS at levels consistent with the in vivo environment, and that the cell line and cell density, as well as the well size and rotation speed, play key roles in yielding spheroids of an appropriate size and quantity. ES-2 shows better adaption to FSS, showing high RCS values, even at low cell densities, and showing increased viability versus static controls, as opposed to the OVCA420. Taken together, these data show how FSS can be easily incorporated into spheroid culture and its potential to be further adapted for future experiments with other cells lines to more fully characterize the effects of FSS on invasion and chemoresistance.

## Figures and Tables

**Figure 1 cells-07-00277-f001:**
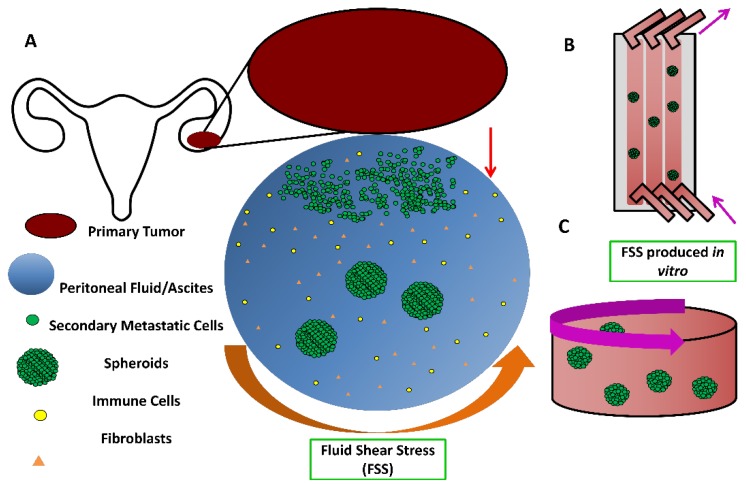
Overview of ovarian cancer metastasis and microenvironment. (**A**) Cells are shed from the ovarian cancer primary tumor into the peritoneal cavity. Following exposure to the fluidic currents and growth factors the cells aggregate into spheroids. This fluid flow can be mimicked in vitro either using microfluidic channels **(B)** or rotary movement of wells **(C).**

**Figure 2 cells-07-00277-f002:**
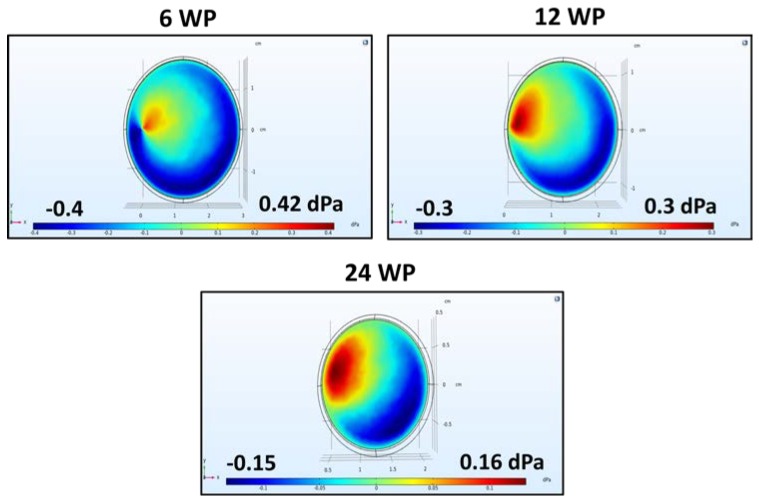
COMSOL simulations of fluid shear stress (FSS) in well plates (WPs) and Petri dishes at 100 rpm. Simulations using the Computational Fluid Dynamics (CFD) module show the tangential shear stress for 6, 12, 24, 48 and 96 WP as well as 5 and 10 cm Petri dishes. Maximum FSS values are reported for each.

**Figure 3 cells-07-00277-f003:**
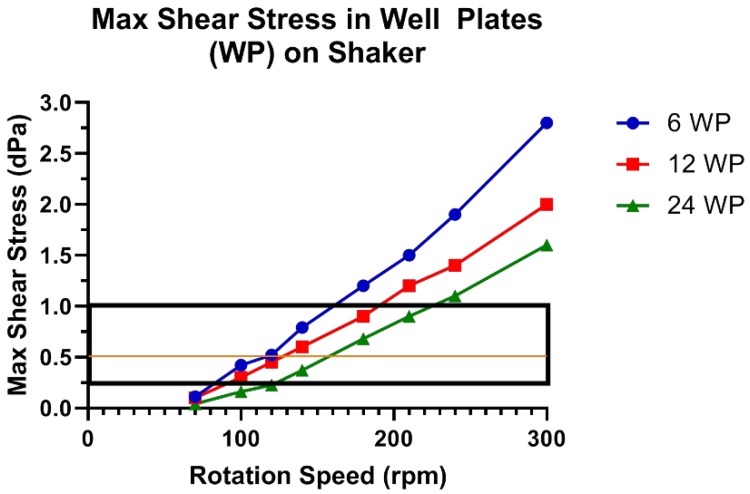
Maximum shear stress in 6, 12 and 24 WP at increasing rotation speeds. While all values are within physiological relevance, 0.25 and 1 decipascals (dPa) were chosen to achieve a low but appreciable value with 0.5 dPa as a rough goal.

**Figure 4 cells-07-00277-f004:**
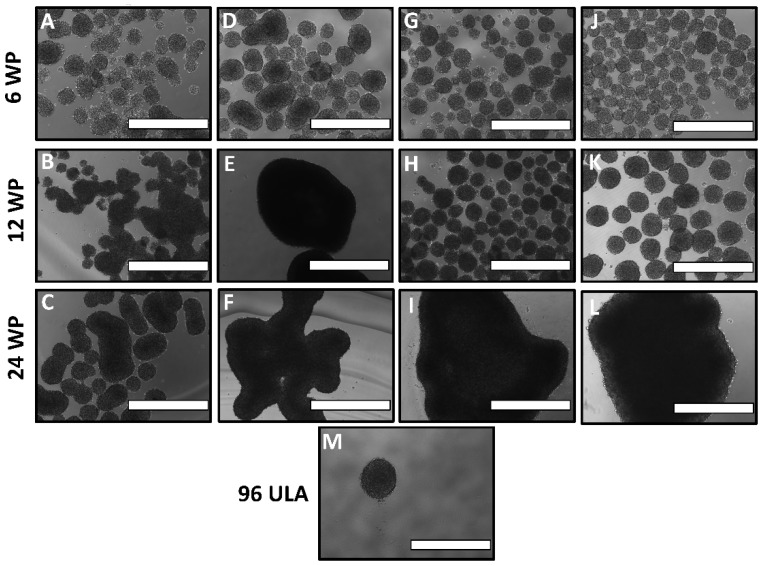
Optimization of well plate size and rotation speed for ES-2 spheroid production. (**A**–**C**) ES-2 spheroids were grown for 72 h in static 6, 12 and 24 WPs, respectively. (**D**–**F**) Spheroids grown under rotating cell culture at 100 rpm for 72 h in 6, 12 and 24 well plates (WPs), respectively. (**G**–**I**) Spheroids grown under rotating cell culture at 120 rpm for 72 h in 6, 12 and 24 WPs, respectively. (**J**–**L**) Spheroids grown under rotating cell culture at 140 rpm for 72 h in 6, 12 and 24 WPs, respectively. (**M**) Static spheroid grown in a round-bottom 96 WP for 72 h. Scale Bar: 1000 μm.

**Figure 5 cells-07-00277-f005:**
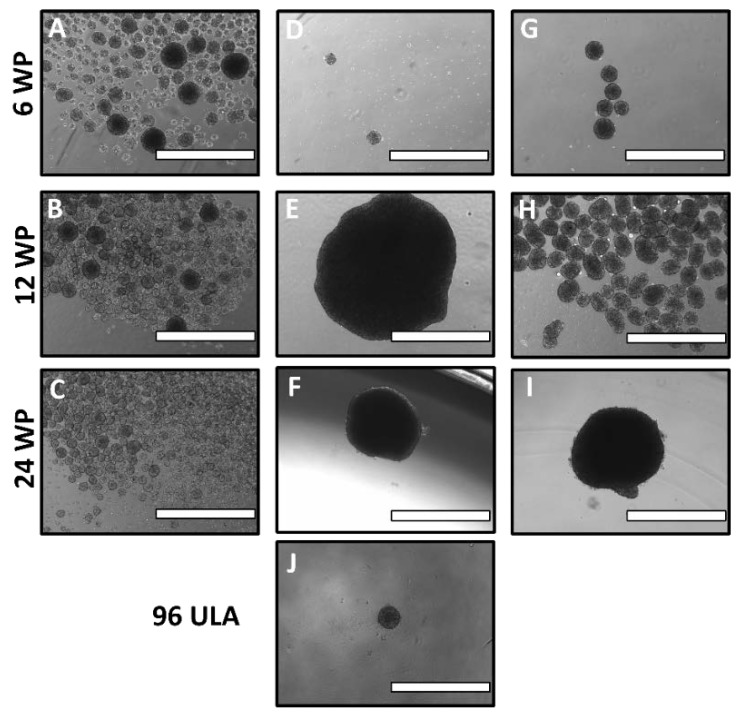
Optimization of well plate size and rotation speed for OVCA420 spheroid production. (**A**–**C**) OVCA420 spheroids were grown for 72 h in static 6, 12 and 24 WPs, respectively. (**D**–**F**) Spheroids grown under rotating cell culture at 120 rpm for 72 h in 6, 12 and 24 WPs, respectively. (**G**–**I**) Spheroids grown under rotating cell culture at 140 rpm for 72 h in 6, 12 and 24 WPs, respectively. (**J**) Static spheroid grown in a round-bottom 96 WP for 72 h. Scale Bar: 1000 μm.

**Figure 6 cells-07-00277-f006:**
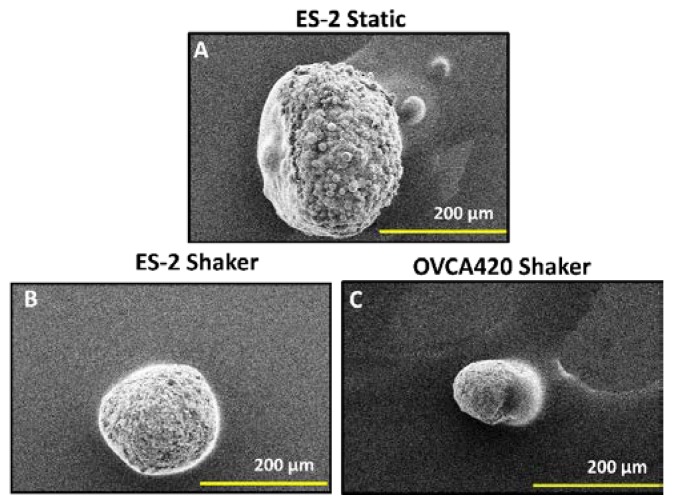
Three-dimensional (3D) nature of shaker-produced spheroids. (**A**) Reference Scanning Electron Microscopy (SEM) image of an ES-2 spheroid generated in 96 RB WP to show 3D structure. (**B**,**C**) SEM images of shaker-produced ES-2—(**B**) and OVCA420 spheroids (**C**) The smooth mass seen on the side of spheroids is mounting media from the coverslip. Scale Bar: 200 μm.

**Figure 7 cells-07-00277-f007:**
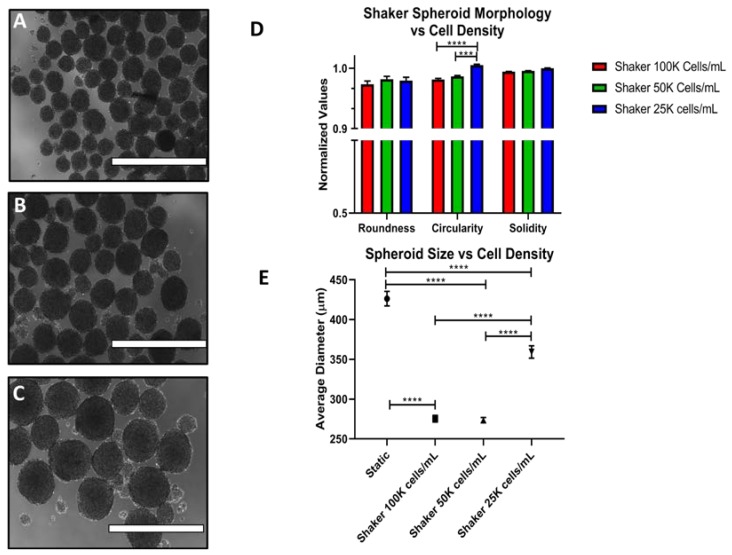
The effects of cell density on ES-2 spheroid properties under shaking conditions. Spheroid characteristics were monitored at 100,000 cells/mL (**A**), 50,000 cells/mL (**B**) and 25,000 cells/mL (**C**) for 72 h. (**D**) Quantification of roundness, circularity and solidity (RCS) values. (**E**) Quantification of spheroid size. Scale Bar: 1000 μm. (*n* = 3).

**Figure 8 cells-07-00277-f008:**
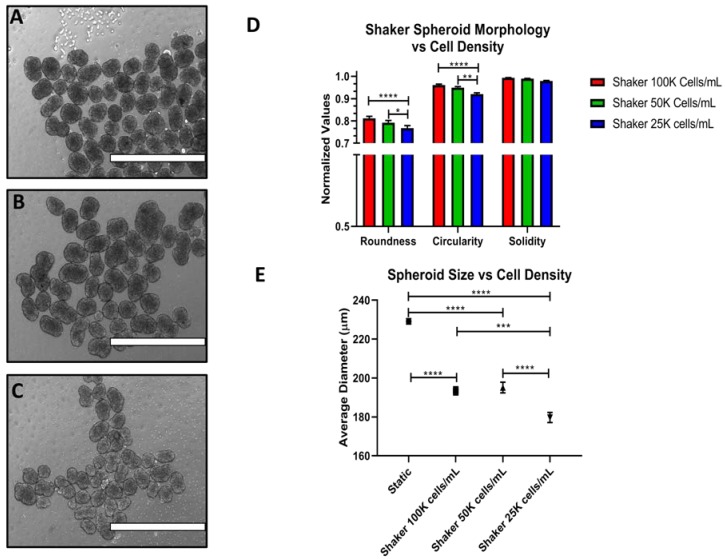
The effects of cell density on OVCA420 spheroid properties under shaking conditions. Spheroid characteristics were monitored at 100,000 cells/mL (**A**), 50,000 cells/mL (**B**) and 25,000 cells/mL (**C**) for 72 h. (**D**) Quantification of RCS values. (**E**) Quantification of spheroid size. Scale Bar: 1000 μm. (*n* = 3).

**Figure 9 cells-07-00277-f009:**
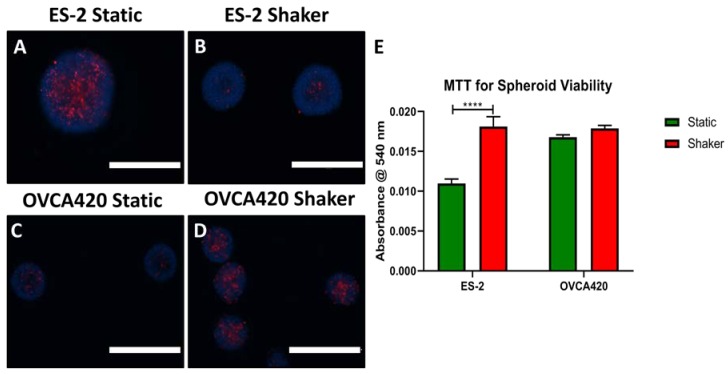
Conservation of spheroid viability following shear stress exposure. Spheroid viability was analyzed in two ways. Cell necrosis was visualized using propidium iodide (red) staining versus Hoechst (blue). Scattered necrotic cells are seen in most spheroids near the center and at the edge under both static (**A**,**B**) and shaker (**C**,**D**) conditions. (**E**) Quantification of spheroid viability using the MTT assay. There was no significant difference in cell viability with both conditions. Scale Bar: 400 μm. (*n* = 3).

**Figure 10 cells-07-00277-f010:**
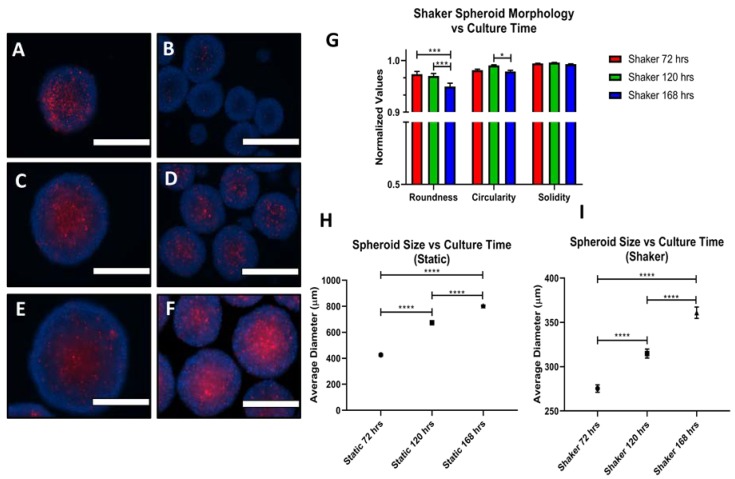
Long-term culture of spheroids on shaker. Propidium iodide (PI) and Hoescht staining was done at 72 (**A**,**B**), 120 (**C**,**D**) and 168 (**E**,**F**) h for static (**A**,**C**,**E**) and shaker (**B**,**D**,**F**) spheroids. (**G**) RSC values from 72–168 h. (**H**,**I**) Spheroid size from 72–168 h for static and shaker spheroids, respectively. Scale Bar: 400 μm. (*n* = 3)

**Table 1 cells-07-00277-t001:** Summary of well size/rotation speed optimization.

Cell Line	Plate Type	Well Diameter (cm)	Rotation Speed (rpm)	* Spheroid Quantity	Morphology	Size Consistency	** Size Relevance
ES-2	6	3.5	0	High	Irregular	Poor	Mixed
	6	3.5	100	High	Irregular	Good	Yes
	6	3.5	120	High	Round	Good	Yes
	6	3.5	140	High	Round	Good	Yes
	12	2.5	0	High	Irregular	Poor	Mixed
	12	2.5	100	Low	Irregular	Good	No
	12	2.5	120	High	Round	Good	Yes
	12	2.5	140	High	Round	Good	Yes
	24	1.8	0	High	Irregular	Poor	Mixed
	24	1.8	100	Low	Irregular	Good	No
	24	1.8	120	Low	Round	Good	No
	24	1.8	140	Low	Round	Good	No
OVCA420	6	3.5	0	High	Round	Poor	Yes
	6	3.5	120	Low	Round	Poor	Yes
	6	3.5	140	Low	Round	Good	Yes
	12	2.5	0	High	Round	Poor	Yes
	12	2.5	120	Low	Round	Good	No
	12	2.5	140	High	Round	Good	Yes
	24	1.8	0	High	Round	Poor	Yes
	24	1.8	120	Low	Round	Good	No
	24	1.8	140	Low	Round	Good	No

* For Spheroid Quantity, “Low” constitutes a condition wherein the number of spheroids per well was ten or fewer. ** For Size Relevance, spheroids above 600 μm in diameter were considered to not be of physiologically-relevant size.

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
