# Peer review of "A Dynamic Culture Method to Produce Ovarian Cancer Spheroids under Physiologically-Relevant Shear Stress"

_cells, 2018, doi:10.3390/cells7120277_

Reviewer 1 Report

This is a well-written and fairly thorough manuscript presenting a new, simple technique for generating multicellular spheroids from ovarian cancer cell lines. A wealth of information supports the importance of spheroid cultures in assessing higher-order aspects of ovarian cancer cell biology (e.g. disaggregation, invasion, drug sensitivity…), but the diverse & non-standardized methodologies for forming spheroids tends to introduce additional, muddying variables to that literature. The authors present a novel technique for spheroid formation using a benchtop multiwell plate shaker. The technique is extremely simple, easily implemented by any lab at very little expense, and is fairly thoroughly characterized for its ability to generate physiologically-relevant size aggregates and for its reproducibility, both of which are impressively high. The authors demonstrate their technique with only two cell lines, which is a very small sampling of those available & utilized in the prevailing literature. However, in this reviewer’s opinion, the spirit of this manuscript is to rather quickly get this technique into the public domain, where it can be validated in a much wider variety of cell lines by many laboratories. In addition, and as the authors acknowledge (in a refreshingly understated manner), this simple technique may very readily be applied to myriad other cell lines from cancers other than ovarian, which increases its potential utility and impact. Thus, while the work offers no new biological insights or mechanisms, it stands to be of considerable utility and technical impact to many researchers in several fields. There are a small number of typographical errors, and the symbols in the graphs (e.g. Fig 7 & 8) could be made larger to increase readability, but short of those very mirror revisions, I feel that the manuscript is acceptable for publication as written.

Author Response

Response to Comments from Reviewer 1

Comment:

This is a well-written and fairly thorough manuscript presenting a new, simple technique for generating multicellular spheroids from ovarian cancer cell lines. A wealth of information supports the importance of spheroid cultures in assessing higher-order aspects of ovarian cancer cell biology (e.g. disaggregation, invasion, drug sensitivity…), but the diverse & non-standardized methodologies for forming spheroids tends to introduce additional, muddying variables to that literature. The authors present a novel technique for spheroid formation using a benchtop multiwell plate shaker. The technique is extremely simple, easily implemented by any lab at very little expense, and is fairly thoroughly characterized for its ability to generate physiologically-relevant size aggregates and for its reproducibility, both of which are impressively high. The authors demonstrate their technique with only two cell lines, which is a very small sampling of those available & utilized in the prevailing literature. However, in this reviewer’s opinion, the spirit of this manuscript is to rather quickly get this technique into the public domain, where it can be validated in a much wider variety of cell lines by many laboratories. In addition, and as the authors acknowledge (in a refreshingly understated manner), this simple technique may very readily be applied to myriad other cell lines from cancers other than ovarian, which increases its potential utility and impact. Thus, while the work offers no new biological insights or mechanisms, it stands to be of considerable utility and technical impact to many researchers in several fields. There are a small number of typographical errors, and the symbols in the graphs (e.g. Fig 7 & 8) could be made larger to increase readability, but short of those very mirror revisions, I feel that the manuscript is acceptable for publication as written.

Response

We thank the reviewer for the effort and time spent in reviewing our manuscript. We appreciate the reviewer’s recommendation of acceptance and understanding of what we seek to achieve with our manuscript. In response to their comment on Figures 7 and 8, the size of Figure 7 is smaller than it was in the original submission, and this may explain why some of the symbols were difficult to read. Size adjustment during the final formatting should resolve all legibility concerns.

Description of Changes to Manuscript

1)  Figure 7’s size has been adjusted to match Figure 8.

Reviewer 2 Report

In this manuscript authors present a dynamic culture method to produce ovarian cancer spheroids and better mimic the physiological conditions in which these cells metastasize following the transcoelomic pathway.  This is a technical work of potential interest for future studies, but some conclusions seem preliminary and in general I believe the results should be taken a bit further with more analysis and cellular assays.

1.       Authors compare their sheer stress method with the commonly used 96 well round bottom static assay, validating their system by obtaining similar results in sphere formation. However, the static 96 well assay is more suitable for high throughput assays. The authors add a physiologically relevant parameter to the assay by allowing the formation of the spheres under sheer stress, but do not show or demonstrate in the manuscript any of the claimed advantages. Do these spheres show an improvement in any quantifiable relevant parameter that demonstrate that the method better mimics what occurs in vivo?

2.       Simulation show pockets of higher FSS in wells. How do they affect the formation of the spheres? Is the formation homogeneous, is there any cell selection or it is completely irrelevant?  This should be discussed further.

3.       Authors test rotation conditions with a large number of cells, and later test different cell densities on selected parameters. Could cell density affect the performance of the different plate and rotation speed used? The initially used of 100000 cells/ml seems very high.

4.       In table 1, cell line data seems to be missing from some cells. Also, although I appreciate the inclusion of the raw data on the supplementary tables, quantitative (mean values for example) and not only qualitative measures should be presented in the table.

5.       To me SEM images add little to the analysis of sphere structure. Moreover, the images presented are not very similar, as claimed. Sphere inclusion on OCT for example and slice staining with relevant cellular markers would be much more informative. Same for the staining presented with PI.

6.       Do necrosis in the core of the spheroids occurs in vivo? A more detailed analysis would be beneficial.

7.       Authors make assumptions on the relationship between their results on their spheroids assays and the metastatic potential of the cell lines used. This is important, as the differences between cell lines should be explained, but analysis of more cell lines is needed to reach conclusions. It seems too preliminary so far.

8.       Authors point to a possible connection between and increase in CSC markers and FSS from previous studies. This is potentially relevant and they should explore it on their spheres. It might give them the physiologically relevant parameter they need to finally endorse their assay.

Minor points:  

1.       How are the images microscopy obtained? This information is missing from the methods.

2.       There are annotations in the figures that are not visible due to font size, for example text under scale bar on Figures 4 and 5. These should be removed or be made visible.

3.       Figure sizes should be similar. Figure 7 is too small to be read properly.

Author Response

Response to Comments from Reviewer 2

Comment:

In this manuscript authors present a dynamic culture method to produce ovarian cancer spheroids and better mimic the physiological conditions in which these cells metastasize following the transcoelomic pathway.  This is a technical work of potential interest for future studies, but some conclusions seem preliminary and in general I believe the results should be taken a bit further with more analysis and cellular assays.

1.      Authors compare their sheer stress method with the commonly used 96 well round bottom static assay, validating their system by obtaining similar results in sphere formation. However, the static 96 well assay is more suitable for high throughput assays. The authors add a physiologically relevant parameter to the assay by allowing the formation of the spheres under sheer stress, but do not show or demonstrate in the manuscript any of the claimed advantages. Do these spheres show an improvement in any quantifiable relevant parameter that demonstrate that the method better mimics what occurs in vivo?

2.      Simulation show pockets of higher FSS in wells. How do they affect the formation of the spheres? Is the formation homogeneous, is there any cell selection or it is completely irrelevant?  This should be discussed further.

3.      Authors test rotation conditions with a large number of cells, and later test different cell densities on selected parameters. Could cell density affect the performance of the different plate and rotation speed used? The initially used of 100000 cells/ml seems very high.

4.      In table 1, cell line data seems to be missing from some cells. Also, although I appreciate the inclusion of the raw data on the supplementary tables, quantitative (mean values for example) and not only qualitative measures should be presented in the table.

5.      To me SEM images add little to the analysis of sphere structure. Moreover, the images presented are not very similar, as claimed. Sphere inclusion on OCT for example and slice staining with relevant cellular markers would be much more informative. Same for the staining presented with PI.

6.      Do necrosis in the core of the spheroids occurs in vivo? A more detailed analysis would be beneficial.

7.      Authors make assumptions on the relationship between their results on their spheroids assays and the metastatic potential of the cell lines used. This is important, as the differences between cell lines should be explained, but analysis of more cell lines is needed to reach conclusions. It seems too preliminary so far.

8.      Authors point to a possible connection between and increase in CSC markers and FSS from previous studies. This is potentially relevant and they should explore it on their spheres. It might give them the physiologically relevant parameter they need to finally endorse their assay.  

Minor points:  

1.  How are the images microscopy obtained? This information is missing from the methods.

2. There are annotations in the figures that are not visible due to font size, for example text under scale bar on Figures 4 and 5. These should be removed or be made visible.

3.  Figure sizes should be similar. Figure 7 is too small to be read properly.

Response

We thank the reviewer for carefully going through our manuscript and offering valuable suggestions. Our responses to the listed concerns are as follows:

1.  Our comparisons include the round-bottom 96 well ULA plate because this is a standard high-throughput assay used for chemoresistance studies (Raghavan, Mehta et al. 2017). Our proposed platform can be used for similar studies in the future and we felt it prudent to compare our system to one that is already utilized.
Also, there is no direct comparison to in vivo spheroids here as it lies outside the scope of our manuscript. Our focus was on development of a platform, and towards that end we have simulated the fluid environment, showing its fluid shear stress (FSS) values are physiologically relevant, and that it can facilitate the formation of ovarian cancer spheroids from two different cell lines, with consistent size and morphology.

We hope to do more extensive biological characterization in our own future studies, and that others will add to that even further. Furthermore, to the best of our knowledge none of the prior studies that have done FSS studies in ovarian cancer spheroids have performed direct comparisons to in vivo spheroids (Hyler, Baudoin et al. 2018), (Ip, Li et al. 2016), (Rizvi, Gurkan et al. 2013), (Avraham-Chakim, Elad et al. 2013), (Li, Ip et al. 2017).

2.  Any system with fluid movement will generate pockets of varying FSS in the way ours does. This is expected physiologically as well, as the peritoneal cavity’s geometry creates areas of varying fluid flow (Pannu and Oliphant 2015). This aides in spheroid formation, as the collection of in vivo spheroids has shown (Deng, Hu et al. 2014). We will also clarify in the manuscript that cell incorporation into spheroids is highly homogenous, showing no clear signs of cell selection.

3.         Our chosen cell density (100,000 cells/mL) for the optimization of well size and rotation speed  was selected because it is within the range of what other publications have used for similar spheroid-formation experiments using rotator plates (Hyler, Baudoin et al. 2018), (Rothermel, Biedermann et al. 2005), (Wigg, Barritt et al. 2009), (Gru, Hohn et al. 1994). Based on our experiments, we believe the spheroid formation dynamics during the aforementioned optimization steps are affected more significantly by the fluid movement, not the cell density.

4.  The reviewer’s point here is well-taken. We have added descriptors that explain our terminology in Table 1 so it is more quantitative.

5.   We believe the purpose of Figure 5 is being misunderstood here. Our SEM images were intended to show the 3D structure of the shaker-produced spheroids, affirming they are 3D as the static spheroids are. This was not meant as a structural comparison. The text has been revised accordingly to ensure our meaning behind these SEM images is clearer. OCT fixation, sectioning and staining, as mentioned by the reviewer, would fall under future directions for further biological characterization of the spheroids.

6.   Necrosis within spheroids is typical when the size grows to be above 300 μm (Hirschhaeuser, Menne et al. 2010). Spheroids that grow in vivo fall within that range, so they are expected to exhibit necrosis at those sizes as other spheroids would (Sodek, Ringuette et al. 2009). A mention of this has been added to the manuscript.

7.  Our statements on the relationship between the metastatic behavior of the cell lines and their spheroid formation are based on several factors.

Firstly, we not only show in Figures 7 and 8 that the ES-2 are forming spheroids more effectively at lower cell densities than OVCA420 but also in Figure 9 that ES-2 spheroids in the shaker show greater viability than the static spheroids while OVCA420 is unchanged. Taken together, these show greater adaptation to FSS for ES-2 than OVCA420, and the more highly metastatic behavior of ES-2 has been previously described (Dier, Shin et al. 2014). This difference makes logical sense, as effective transcoelomic metastasis would be greatly facilitated by prompt cell adaptation to FSS.

Secondly, this trend was also observed by Hyler et al., using mouse cell lines of varying metastatic potential (Hyler, Baudoin et al. 2018), so that trend has been replicated here in human cell lines.
Our goal with this manuscript is to create a platform, and our observations here of the differential behavior between ES-2 and OVCA420 are meant to serve as guides for other labs that wish to adapt this method, to facilitate predictions of how their own cell lines may behave under these conditions. The reviewer’s point is understood, and this trend is certainly one that warrants further investigation with additional cell lines, but we feel our observations with the chosen cell lines are valid for our purposes of developing a platform.

8.  As mentioned previously, biological characterization lies outside the scope of our manuscript. Our emphasis is on developing a platform and a method that is widely applicable for many labs to do biological characterization on such factors as epithelial-to-mesenchymal transition (EMT) and cancer stem cell (CSC) markers.

Minor points:

1.      The text has been edited to have a specific section on the microscopy techniques used here.

2.      A reference for the scale bars in the images in Figures 4 and 5 is defined at the end of the figure legend.

3.      Figure 7’s size has been adjusted to match the size of Figure 8.

Description of Changes to Manuscript

1)     On line 2, the capitalization in the title has been adjusted.

2)    The email address of James Castracane has been corrected on line 9.

3)     The equivalence of dynes/cm2 and dPa is clarified on line 75.

4)     On line 118, the source of our cell lines has been more clearly identified.

5)     An additional section on our microscopy techniques has been added on line 136, and the following subsection headings have been updated accordingly

6)     The high incorporation of cells into spheroids has been added on line 217.

7)      Figure 5 has been centered and its size adjusted.

8)      On lines 251 and 253, descriptors for the spheroid quantity and size relevance have been added to Table 1, as well as markers tying them to the table labels.

9)     Lines 255-258, the text has been edited to clarify the intent of the SEM images to show 3D structure.

10)   Figure 6’s legend has been edited to reflect the changes mentioned above.

11)    Figure 7’s size has been adjusted to match Figure 8.

12)    On Line 299, the meaning of necrosis has been clarified.

13)    On line 301, the appearance of spheroid necrosis at certain sizes has been explicitly stated.

14)    Figure 9 has been enlarged.

15)     Figure 10 has been centered.

References

Avraham-Chakim, L., D. Elad, U. Zaretsky, Y. Kloog, A. Jaffa and D. Grisaru (2013). "Fluid-flow induced wall shear stress and epithelial ovarian cancer peritoneal spreading." PLoS One 8(4): e60965.

Deng, X., J. Hu, M. J. Cunningham and E. Friedman (2014). "Mirk kinase inhibition targets ovarian cancer ascites." Genes & cancer 5(5-6): 201.

Dier, U., D. H. Shin, L. P. Hemachandra, L. M. Uusitalo and N. Hempel (2014). "Bioenergetic analysis of ovarian cancer cell lines: profiling of histological subtypes and identification of a mitochondria-defective cell line." PLoS One 9(5): e98479.

Gru, R., H.-P. Hohn, M. Mareel and H.-W. Denker (1994). "Adhesion and invasion of three human choriocarcinoma cell lines into human endometrium in a three-dimensional organ culture system." Placenta 15(4): 411-429.

Hirschhaeuser, F., H. Menne, C. Dittfeld, J. West, W. Mueller-Klieser and L. A. Kunz-Schughart (2010). "Multicellular tumor spheroids: an underestimated tool is catching up again." J Biotechnol 148(1): 3-15.

Hyler, A. R., N. C. Baudoin, M. S. Brown, M. A. Stremler, D. Cimini, R. V. Davalos and E. M. Schmelz (2018). "Fluid shear stress impacts ovarian cancer cell viability, subcellular organization, and promotes genomic instability." PLoS One 13(3): e0194170.

Ip, C. K., S. S. Li, M. Y. Tang, S. K. Sy, Y. Ren, H. C. Shum and A. S. Wong (2016). "Stemness and chemoresistance in epithelial ovarian carcinoma cells under shear stress." Sci Rep 6: 26788.

Li, S. S., C. K. Ip, M. Y. Tang, S. K. Sy, S. Yung, T. M. Chan, M. Yang, H. C. Shum and A. S. Wong (2017). "Modeling Ovarian Cancer Multicellular Spheroid Behavior in a Dynamic 3D Peritoneal Microdevice." J Vis Exp(120).

Pannu, H. K. and M. Oliphant (2015). "The subperitoneal space and peritoneal cavity: basic concepts." Abdom Imaging 40(7): 2710-2722.

Raghavan, S., P. Mehta, M. R. Ward, M. E. Bregenzer, E. M. Fleck, L. Tan, K. McLean, R. J. Buckanovich and G. Mehta (2017). "Personalized medicine–based approach to model patterns of chemoresistance and tumor recurrence using ovarian cancer stem cell spheroids." Clinical Cancer Research.

Rizvi, I., U. A. Gurkan, S. Tasoglu, N. Alagic, J. P. Celli, L. B. Mensah, Z. Mai, U. Demirci and T. Hasan (2013). "Flow induces epithelial-mesenchymal transition, cellular heterogeneity and biomarker modulation in 3D ovarian cancer nodules." Proceedings of the National Academy of Sciences 110(22): E1974-E1983.

Rothermel, A., T. Biedermann, W. Weigel, R. Kurz, M. Rüffer, P. G. Layer and A. A. Robitzki (2005). "Artificial design of three-dimensional retina-like tissue from dissociated cells of the mammalian retina by rotation-mediated cell aggregation." Tissue engineering 11(11-12): 1749-1756.

Sodek, K. L., M. J. Ringuette and T. J. Brown (2009). "Compact spheroid formation by ovarian cancer cells is associated with contractile behavior and an invasive phenotype." Int J Cancer 124(9): 2060-2070.

Wigg, A. J., G. J. Barritt, G. P. Young and J. W. Phillips (2009). "Inhibition of oxidative stress and apoptosis enables extended maintenance of integrity and function of isolated hepatocytes in suspension." Journal of gastroenterology and hepatology 24(6): 1082-1088.

Round  2

Reviewer 2 Report

I appreciate the effort in replying to this reviewers’ concerns. I still believe the work would greatly benefit for some more simple biological testing to better reflect the improvement of the technique, as for example the use of another cell line in the assays or some improvement on the imaging used. However, I understand the explanation of the authors on the objective of the manuscript as a more technical document and look forward to other works with more biological insights. Some minor comments:

Even with the changes, care should be taken on the size of figures and legends on the final edited document. I see the scale bar reference at the end of the legend in Figures 4 and 5, but still the elimination of non-readable text in the images will provide more elegance and clarity to the figures.

Author Response

Response to Comments from Reviewer 2

Comment

I appreciate the effort in replying to this reviewers’ concerns. I still believe the work would greatly benefit for some more simple biological testing to better reflect the improvement of the technique, as for example the use of another cell line in the assays or some improvement on the imaging used. However, I understand the explanation of the authors on the objective of the manuscript as a more technical document and look forward to other works with more biological insights. Some minor comments:

Even with the changes, care should be taken on the size of figures and legends on the final edited document. I see the scale bar reference at the end of the legend in Figures 4 and 5, but still the elimination of non-readable text in the images will provide more elegance and clarity to the figures.

Response

We thank the reviewer for their input and understanding. We have added thicker, more visible scale bars to the figures so they are more visible and there is no non-readable text on them. The reviewer specifically mentioned Figures 4 and 5 but for consistency those scale bars were added to other figures with brightfield or fluorescence images.

Description of Overall Changes to Manuscript

1)            Figures 4-5 and 7-10 have been replaced with ones that have new, thicker scale bars with no accompanying text.